# Air Quality Index (AQI) Did Not Improve during the COVID-19 Lockdown in Shanghai, China, in 2022, Based on Ground and TROPOMI Observations

Qihan Ma [1], Jianbo Wang [1], Ming Xiong [2] and Liye Zhu [1,3,4,*]

1 School of Atmospheric Sciences, Sun Yat-Sen University, Zhuhai 519082, China
2 School of Computer Science and Engineering, Sun Yat-Sen University, Guangzhou 510006, China
3 Southern Marine Science and Engineering Guangdong Laboratory (Zhuhai), Zhuhai 519082, China
4 Key Laboratory of Tropical Atmosphere-Ocean System, Ministry of Education, Zhuhai 519082, China
* Correspondence: zhuly37@mail.sysu.edu.cn

**Abstract:** The lockdowns from the coronavirus disease of 2019 (COVID-19) have led to a reduction in anthropogenic activities and have hence reduced primary air pollutant emissions, which were reported to have helped air quality improvements. However, air quality expressed by the air quality index (AQI) did not improve in Shanghai, China, during the COVID-19 outbreak in the spring of 2022. To better understand the reason, we investigated the variations of nitrogen dioxide ($NO_2$), ozone ($O_3$), $PM_{2.5}$ (particular matter with an aerodynamic diameter of less than 2.5 μm), and $PM_{10}$ (particular matter with an aerodynamic diameter of less than 10 μm) by using in situ and satellite measurements from 1 March to 31 June 2022 (pre-, full-, partial-, and post-lockdown periods). The results show that the benefit of the significantly decreased ground-level $PM_{2.5}$, $PM_{10}$, and $NO_2$ was offset by amplified $O_3$ pollution, therefore leading to the increased AQI. According to the backward trajectory analyses and multiple linear regression (MLR) model, the anthropogenic emissions dominated the observed changes in air pollutants during the full-lockdown period relative to previous years (2019–2021), whereas the long-range transport and local meteorological parameters (temperature, air pressure, wind speed, relative humidity, and precipitation) influenced little. We further identified the chemical mechanism that caused the increase in $O_3$ concentration. The amplified $O_3$ pollution during the full-lockdown period was caused by the reduction in anthropogenic nitrogen oxides ($NO_x$) under a VOC-limited regime and high background $O_3$ concentrations owing to seasonal variations. In addition, we found that in the downtown area, ground-level $PM_{2.5}$, $PM_{10}$, and $NO_2$ more sensitively responded to the changes in lockdown measures than they did in the suburbs. These findings provide new insights into the impact of emission control restrictions on air quality and have implications for air pollution control in the future.

**Keywords:** COVID-19; Shanghai; air quality; $O_3$ pollution; TROPOMI

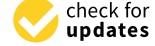



## 1. Introduction

In December 2019, the abrupt outbreak of COVID-19 produced previously unseen societal impacts in China [1]. The city of Shanghai, as the first echelon, was placed under lockdown and officially launched the First-Level Public Health Emergency Response to epidemic prevention and control. The tremendous epidemic in Shanghai was controlled in March 2020. Since then, Shanghai has had only a small number of occasionally imported cases or localized outbreaks on a small scale, which had been properly handled and controlled [2], and there was no second wave of tremendous outbreak. However, in early March 2022, a new round of COVID-19, caused by the spread of the Omicron variant, rapidly attacked the whole city of Shanghai, which led to the second lockdown management, from March 28 to June 1, 2022. Consequently, human mobility, public transportation options, manufacturing, and economic growth were greatly reduced. For instance, in the

first half of 2022, Shanghai's gross domestic product (GDP) decreased by 5.7% relative to the previous year [3].

There is no doubt that the COVID-19 epidemic is a tragedy and has caused an adverse impact on people's lifestyles and production activities [4–7]. Nonetheless, it has provided an unprecedented experiment in which substantial reductions have occurred in anthropogenic activities, which are unique opportunities to assess the efficiency of air pollution mitigation. The AQI describes the degree of air cleanliness and its impact on health. In terms of the AQI, many studies have suggested that air quality generally improved at multiple spatial scales throughout the world during the lockdowns [8–12]. Focusing on China, Wang et al. [13] reported that the AQI averaged over 366 urban areas decreased by 20% ($-28$ points) during the historical wintertime COVID-19 lockdowns (hereafter referred to as HWCL, approximately from late January 2020 to March 2020, in China) as compared with that before the control period, which was due mainly to the substantial reduced primary emissions from anthropogenic activities.

Despite the widely distributed air quality improvements and decreases in primary emissions, it has been reported that several air pollution events still existed in some regions of China during the HWCL [14–19]. The substantial declines in $NO_x$ emissions alleviated lower $O_3$ titration in urban areas of China, leading to amplified $O_3$ pollution [20]. Moreover, previous studies have shown that in eastern China, increases in $O_3$ have resulted in an increased atmospheric oxidizing capacity and therefore facilitated the formation of secondary particular matter (PM), which has caused several heavy winter haze events [19]. Additionally, meteorology has a non-negligible impact on the formation or diffusion of air pollutants [21], and unfavorable meteorological conditions may lead to severe air pollution events [22]. For instance, several severe $PM_{2.5}$ pollution episodes were simultaneously observed in northern China during the HWCL [14–18]. This was attributable mostly to unfavorable meteorological conditions, including low wind speeds and increased relative humidity (RH). As a result, under certain circumstances, the emission reduction of anthropogenic sources cannot completely offset the adverse impact of secondary aerosol formation or unfavorable meteorological conditions on air quality [14–19]. According to these understandings, we cannot infer that air quality will improve during this springtime lockdown in Shanghai. Additionally, air pollutants usually present spatial heterogeneity, which is related to geographical location, city development, and the emission sectors [23,24]. The national and global trends of air pollutants have been quantified in detail during the COVID-19 epidemic [24,25], whereas research on the changes in urban-scale and suburb-scale air pollutants in response to the lockdown measures is still limited. Therefore, all these should be addressed to achieve the objectives of regional air quality management.

The AQI and five representative air pollutants, specifically $PM_{2.5}$, $PM_{10}$, $NO_2$, $O_3$, and formaldehyde (HCHO), were selected for an analysis of the impact before and after the lockdown. $PM_{2.5}$ and $PM_{10}$ have an important relationship with human premature death from cardiovascular and respiratory diseases [1]. $NO_2$ is usually regarded as a direct indicator for evaluating industrial pollution, vehicle exhaust emissions, and biomass burning [21,24], which participate in many chemical reactions and crucially affect air quality. $O_3$ pollution is one of the main air quality challenges in China [26,27], which damages human respiratory and immune systems and causes related diseases [28]. Anthropogenic $NO_x$ and volatile organic compounds (VOCs) are the main precursors of $O_3$ and generate $O_3$ through a series of photochemical reactions [29]. Regarded as a proxy for VOCs reactivity, HCHO is an important intermediate in the oxidative degradation of atmospheric VOCs [29,30]. The ratio of HCHO to $NO_2$ ($HCHO/NO_2$, FNR) is assumed as an indicator of the relative sensitivity of surface $O_3$ to $NO_x$ and VOCs emissions (details are in Text S2) [31]. The formation and the changed mechanism of $O_3$ under major events were widely analyzed by using the estimated FNR (e.g., 2008 Beijing Olympics, 2014 Asia-Pacific Economic Cooperation) [32,33]. Thus, first understanding the formation mechanism of regional $O_3$ is an important prerequisite for scientific $O_3$ control. Furthermore, the complex chemical processes of different air pollutants also significantly impact air quality [34].

In this study, we utilized both ground-based and high-resolution satellite data to ensure the reliability and robustness of the results. The following four aspects are taken as specific objectives: (1) analyzing the general variations of different air pollutants and the AQI during different periods of the COVID-19 pandemic in Shanghai; (2) investigating the effect of long-range transport and quantifying the meteorologically and anthropogenically driven changes in ground-level $NO_2$, $O_3$, $PM_{2.5}$, and $PM_{10}$ concentrations during the full-lockdown period; (3) exploring the driving force of the unexpected increase in the AQI and interpreting the relevant formation mechanism of $O_3$ over Shanghai on the basis of using FNR; and (4) investigating the differences in changes in ground-level $NO_2$, $O_3$, $PM_{2.5}$, and $PM_{10}$ in response to the COVID-19 lockdown measures between downtown and the suburbs. It is expected that through the new round of the COVID-19 epidemic in Shanghai, the results are helpful in gaining new insights into air pollution control strategies.

## 2. Methodology

### 2.1. Study Region and Time

Shanghai is located between 30°40′–31°53′N and 120°52′–122°12′E on the east of the Asian continent and the front edge of the Yangtze River Delta (YRD) (Figure 1a). Shanghai occupied a total area of 6340 km² and had a resident population of almost 24.87 million by the end of 2020 [35]. It is an important international economic, financial, trade, and shipping center in China. Shanghai is composed of 16 municipal districts. The downtown includes the Huangpu (HP), Xuhui (XH), Changning (CN), Jing'an (JA), Putuo (PT), Hongkou (HK), and Yangpu (YP) districts, and it is an important comprehensive transportation hub and a modern service industry development center. Although the total area of downtown is 664 km², its resident population accounts for approximately half of that in Shanghai. The suburban area consists of the Minhang (MH), Baoshan (BS), Jiading (JD), Jinshan (JS), Songjiang (SJ), Qingpu (QP), Fengxian (FX), Chongming (CM), and semisuburb Pudong New Area (PD) districts (Figure 1b).

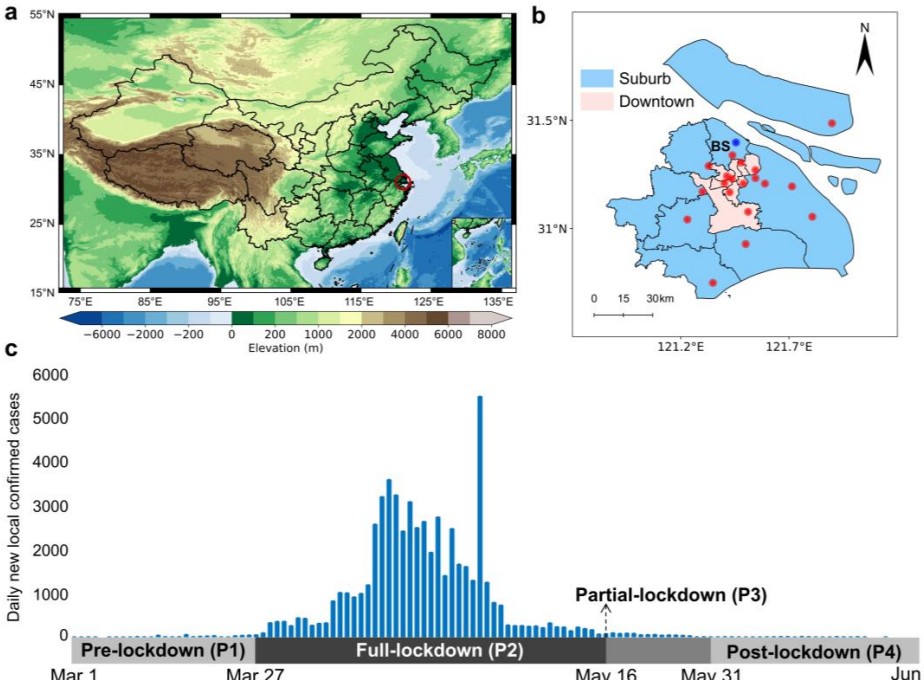

**Figure 1.** (**a**) The topography of China. The red circle represents the location of Shanghai; (**b**) suburbs and downtown Shanghai. Bold black text indicates the location of the BS district, red dots represent the 19 air quality monitoring stations, and the blue dot represents the BS meteorological station; (**c**) the study time series of daily COVID-19 new local confirmed cases from March 1 to June 31 (i.e., from P1 to P4).

The 4-month study period from March 1 to June 31, 2022, is divided into four periods according to the development of the COVID-19 epidemic and government management measures (Figure 1c). Because Shanghai was officially blocked on 28 March 2022, the first period (P1) is defined as the prelockdown stage, from March 1 to 27. The second period (P2), from March 28 to May 16, is defined as the full-lockdown stage. The third period (P3) is May 17 to 31, which is the partial-lockdown stage. The epidemic situation in P2 remained the grimmest, where all kinds of anthropogenic activities were substantially reduced and with 56,850 COVID-19 local confirmed cases in total (Figure 1c) [36]. On 17 May of P3, the Shanghai Municipal Health Commission officially announced that all 16 districts eliminated COVID-19 outside of the quarantine zones [37]. While stay-at-home protection and staggered-shift policies were still implemented, gradual work resumption was taking place across some major projects (e.g., transportation and factory operation). Several supermarkets, restaurants, pharmacies, and other public services reopened. The fourth period (P4) is June 1 to 31 and represents the post-lockdown stage. The Shanghai Municipal People's Government announced that from 1 June, residents would regain access to residential areas and all public transportation would be restored, except for several medium- and high-risk controlled areas [38].

*2.2. Data Sources*

2.2.1. Air Quality Data and Meteorological Data

In this study, the in situ observations of the AQI; the daily averaged concentrations of $PM_{2.5}$, $PM_{10}$, and $NO_2$; the daily maximum 8-hour average (MDA8) $O_3$ concentrations; and the daily dominant pollutant datasets were obtained from the Shanghai Municipal Ecological Environment Bureau, including 19 air quality monitoring stations in 16 districts of Shanghai (Figure 1b). In China, the AQI is defined as the highest of the individual air quality index (IAQI) values (definitions are in Text S1). An AQI value less than 100 indicates good air quality, according to the Technical Regulation on Ambient Air Quality Index of China [39]. The daily dominant pollutant has the highest IAQI value, which determines the daily AQI (Text S1). In addition, we investigated the influence of local meteorological factors and anthropogenic activities. Meteorological parameters, including air temperature, atmospheric pressure, wind speed, RH, and precipitation, measured at the BS meteorological station (Figure 1b), were collected and sorted by the National Climatic Data Center and the UK Meteorological Office. The information on the observed data is summarized in Table S1. The meteorological parameters were processed into daily averages at the same time resolution as air pollutants. The BS meteorological station is not far from the BS monitoring station. The distance between them is approximately 8 km (Figure 1b), ensuring the distribution consistency and reliability of the case-study results.

We have adopted a comparative approach to explore the impact of the COVID-19 epidemic on air quality from P1 to P4. All the data from previous years (2019–2021) during the same periods of 2022 were also analyzed as historical references. The Wilcoxon signed rank test (details are in Text S3) was used to analyze whether there were significant changes in different air pollutants and the AQI. This test was selected because the data during each research period were not required to be normally distributed (Text S3 and Table S2).

2.2.2. Sentinel-5P/TROPOMI $NO_2$ and HCHO Data

The Sentinel-5 Precursor (S5P) is a low-Earth-orbit polar and polar sun-synchronous satellite that launched on October 13, 2017, with an ascending node (from south to north) that crosses the equator at around 13:30 local solar time [40]. The function of S5P is to provide information on and services for air quality, climate, and the ozone layer [41]. The tropospheric monitoring instrument (TROPOMI) is the single payload of the S5P spacecraft, which can effectively monitor trace gas components, including $O_3$, $NO_2$, $SO_2$, CO, $CH_4$, HCHO, and aerosol properties, in the atmosphere around the world [41]. As the most technologically advanced instrument with the highest spatial resolution, TROPOMI provides near-daily global coverage and an approximately 2600 km wide imaging swath [41].

In terms of the validation of the TROPOMI $NO_2$ and HCHO tropospheric VCD data, much work has been conducted by using aircraft-borne profile measurements, ground-based observations, and model simulations [42–46]. These studies have generally demonstrated the good quality of the TROPOMI tropospheric column data, which are suited to a variety of applications, although they have some negative biases compared with in situ measurements (details are in Text S4). Moreover, Sentinel-5P/TROPOMI provides more-adequate spatial resolution and observation information on air pollutants in the troposphere, so as to cope with the COVID-19 epidemic and fill the gap from the low spatial distribution limit of ground monitoring [24,47,48]. TROPOMI observations have been widely utilized by previous studies to monitor the air quality and analyze the variations of $NO_2$ and HCHO during the HWCL around the world [48–50].

In this study, we used the TROPOMI tropospheric level 2 (L2) $NO_2$ and HCHO OFFL (offline) product data (from March 1 to June 1 in 2020 and in 2021), which were obtained from the Copernicus Open Access Hub. Detailed descriptions of the $NO_2$ and HCHO algorithms can refer to the S5P product ATBD (algorithm theoretical basis document) [51,52]. The near-ground pixel size of $NO_2$ and HCHO products is $7.0 \times 3.5$ km$^2$, while it was revised to $5.5 \times 3.5$ km$^2$ on 6 August 2019. For the TROPOMI data, it is important to conduct data quality control before analysis [53,54]. A quality assurance value (qa_value) is a continuous flag variable that ranges from 0 (no output) to 1 (all is well), indicating the status and quality of the retrieval result. Each ground pixel of $NO_2$ and HCHO is selected with a qa_value > 0.5 and a cloud radiation fraction > 0.5 (details are in Text S4), which exclude the cloud-covered scenes (i.e., cloud radiance fraction > 0.5), errors, and problematic retrievals [53,54]. We distributed TROPOMI observations on a two-dimensional latitude–longitude gridded map with a daily temporal resolution and a spatial resolution of $0.1° \times 0.1°$. The daily averaged tropospheric $NO_2$ and HCHO vertical column densities (*VCDs*) in each grid were calculated as follows:

$$\overline{VCDs}_{i,j,y,m,d} = \begin{cases} \frac{1}{n}\sum_1^n VCDs_{i,j,y,m,d,n}, \ if \ n \neq 0 \\ 0 \ , \ if \ n = 0 \end{cases} \tag{1}$$

where *i* is the index of latitude (*i* is from 1 to 1800), *j* is the index of longitude (*j* is from 1 to 3600), *y* is the year, *m* is the month, *d* is the day, and *n* refers to the number of pixels in the grid (*i*, *j*) on the day (*y*, *m*, *d*) (details are in Text S4). Subsequently, these TROPOMI daily data were averaged in each period from P1 to P4.

### 2.3. Multiple Linear Regression Model

The MLR model establishes a quantitative statistical relationship between a forecast quantity and multiple variables, which has been widely applied to meteorological statistics and forecasts. Meteorologically driven changes in regional-scale $PM_{2.5}$ and $O_3$ concentrations are comprehensively understood by utilizing the MLR model [55–57]. However, most MLR models in previous studies that have accounted for the impact of seasonal variations or long-term trends were not applicable to the short-time interval of the COVID-19 lockdown (i.e., P2, 50 days) at the city scale [55–57]. Recently, Fu et al. [58] developed an MLR model to quantify meteorological and anthropogenic influences on $O_3$ during the COVID-19 lockdown period in Guangxi, China. On the basis of using the method in Fu et al. [58], we further developed a stepwise MLR model to explore the effect of meteorologically and anthropogenically driven changes on $NO_2$, $O_3$, $PM_{2.5}$, and $PM_{10}$ concentrations during the new round of COVID-19 lockdowns in Shanghai, using 4-year (2019–2022) ground-based monitoring data from the BS site.

The short-term variations in air pollutants are significantly affected by local weather systems and short-term fluctuations in primary emissions [59]. In this study, we included five meteorological variables (temperature, pressure, RH, wind speed, and precipitation) in the stepwise MLR model. To minimize the influences of correlations between predictors,

we first tested the multicollinearity among these variables via a variance inflation factor (VIF) analysis [57]. VIF is calculated as follows:

$$\text{VIF}_i = \frac{1}{1 - R_i^2} \tag{2}$$

where $R_i^2$ represents the multiple coefficient of determination of the i-th independent variable regressed on all the remaining variables. We set the threshold at which the variable with a VIF value less than 10 would be accepted; otherwise, it would be removed. All the variables that we selected as predictors were within the tolerance of multicollinearity (Table S3).

Stepwise regression obtains the best model fit by adding predictors with significant contributions and deleting insignificant predictors. The basic form trained from historical data is as follows:

$$C_{0i}(t_0) = \beta_{i,0} + \sum_{k=1}^{N} \beta_{i,k} \times M_{0k}(t_0) + \varepsilon \tag{3}$$

where $C_{0i}(t_0)$ is the observed daily concentration of air pollutant i (e.g., $NO_2$) from 2019 to 2021 during P2 for the BS site, $M_{0k}(t_0)$ presents the k-th daily meteorological element, $\beta_{i,0}$ is the constant term of the regression equation (i.e., intercept), $\beta_{i,k}$ is the regression coefficient for the k-th meteorological variable, and $\varepsilon$ is the residual term. From 28 March to 16 May 2020 (P2), the previous COVID-19 epidemic in Shanghai had ended and normal human production and a normal way of life had resumed. Therefore, the data from 2020 can also present the characteristic of no sharp fluctuations in anthropogenic influence (i.e., constant anthropogenic influence) as that in 2019 and 2021. The stepwise MLR model guarantees that each variable selected for i-th air pollutant in the equation is significant (Table S4). The regression coefficients of nonexistent predictors in the final MLR model were set as 0. Furthermore, to improve the accuracy of the results, we removed the outliers that were more than or less than the three times the standard deviations [58]. The adjusted coefficient of determination ($aR^2$) for the regression equation reflects the reliability of the MLR model to explain the variations of the air pollutants during P2 (Table S4). In the study, the $aR^2$ values for $NO_2$, $O_3$, $PM_{2.5}$, and $PM_{10}$ are 0.3, 0.5, 0.2, and 0.4, respectively, which are within a reasonable range, indicating that the model has good explanatory and predictive abilities [55,57].

Then the i-th meteorologically driven air pollutant concentrations in 2022 are predicted by $p_i(t)$ and given by Equation (4). $\Delta C_{Mi}(t)$ in Equation (5) presents the meteorologically driven change in the i-th air pollutant concentrations of 2022 compared to previous years (2019–2021). The difference between the observed value ($\Delta C_i(t)$) and $\Delta C_{Mi}(t)$ is defined as nonmeteorologically driven change ($\Delta C_{Ai}(t)$), which is attributed mainly to anthropogenic activities according to Equation (6) [55,57]. The relevant equations are as follows:

$$p_i(t) = \beta_{i,0} + \sum_{k=1}^{N} \beta_{i,k} \times M_k(t) + \varepsilon \tag{4}$$

$$\Delta C_{Mi}(t) = \sum_{k=1}^{N} \beta_{i,k} \times \Delta M_k \tag{5}$$

$$\Delta C_{Ai}(t) = \Delta C_i(t) - \Delta C_{Mi}(t) \tag{6}$$

where $\Delta M_k$ is the anomalies in the k-th meteorological predictor of 2022 compared with that from 2019 to 2021 and $\Delta C_i(t)$ is the difference in the observed i-th air pollutant concentration of 2022 in respect to the previous 3-year baseline.

*2.4. Backward Trajectory Simulation*

To determine the origin of air masses and establish source–receptor relationships in Shanghai, we used the hybrid single-particle Lagrangian integrated trajectory (HYSPLIT) model from the National Oceanic and Atmospheric Administration's (NOAA) Air Resources Laboratory (ARL), which enabled us to simulate the backward trajectories. With the input meteorological data from the National Center for Environmental Prediction



(NCEP)/Global Data Association System (GDAS) at a $1° \times 1°$ horizontal resolution, the 24-hour backward trajectories were computed at 1-hour interval at a height of 500 m above the ground level from P1 to P4 in 2022. On the basis of the HYSPLIT cluster analysis, all backward trajectories were categorized into distinct transport patterns.

## 3. Results

### 3.1. General Variations of Air Pollution between Different Periods

3.1.1. Variations of Ground-Observed Ambient Air Pollutants and AQI

Figure 2 presents a time series of the daily $NO_2$, $PM_{2.5}$, and $PM_{10}$ concentrations; the daily MDA8 $O_3$ concentrations; and the daily AQI, which is based on in situ measurements over Shanghai from P1 to P4 in 2022. The timing of the decrease in many air pollutants ($NO_2$, $PM_{2.5}$, and $PM_{10}$) was remarkably coincident with the lockdown measurements over Shanghai from P1 to P2 in 2022, while $O_3$ concentrations and the AQI increased. During P2 of 2022, $PM_{2.5}$ concentrations declined by 29.8% ($-9.7$ μg/m$^3$) and showed a high significant difference ($p < 0.01$) compared with the same period in the previous year (Figure 3 and Table S5). Similarly, $PM_{10}$ also presented decreasing trend by 39.3% ($-23.6$ μg/m$^3$) and a high significant difference ($p < 0.01$). The $PM_{2.5}$ concentrations during P2 of 2022 ranged from 9 μg/m$^3$ to 50 μg/m$^3$, with an average value of 22.9 μg/m$^3$, thus meeting the level II standard of the Environmental Air Quality Standard (EAQS) of China (GB3095-2012) (Table S6) [60]. Moreover, the $PM_{10}$ concentrations during P2 ranged from 9 μg/m$^3$ to 90 μg/m$^3$, with an average value of 36.4 μg/m$^3$, which met the level I standard of EAQS. As a result, the pollution of $PM_{2.5}$ and $PM_{10}$ had been significantly reduced.

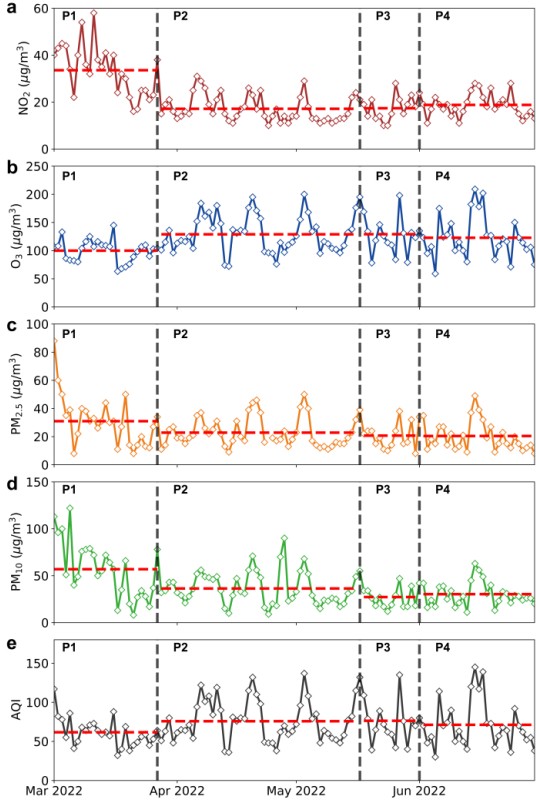

**Figure 2.** Time series of daily average $NO_2$ (**a**), $PM_{2.5}$ (**b**), $PM_{10}$ (**c**), MDA8 $O_3$ concentrations (**d**), and the daily AQI (**e**), based on in situ measurements over Shanghai from P1 to P4 in 2022. The dashed red line indicates the mean value in each period.

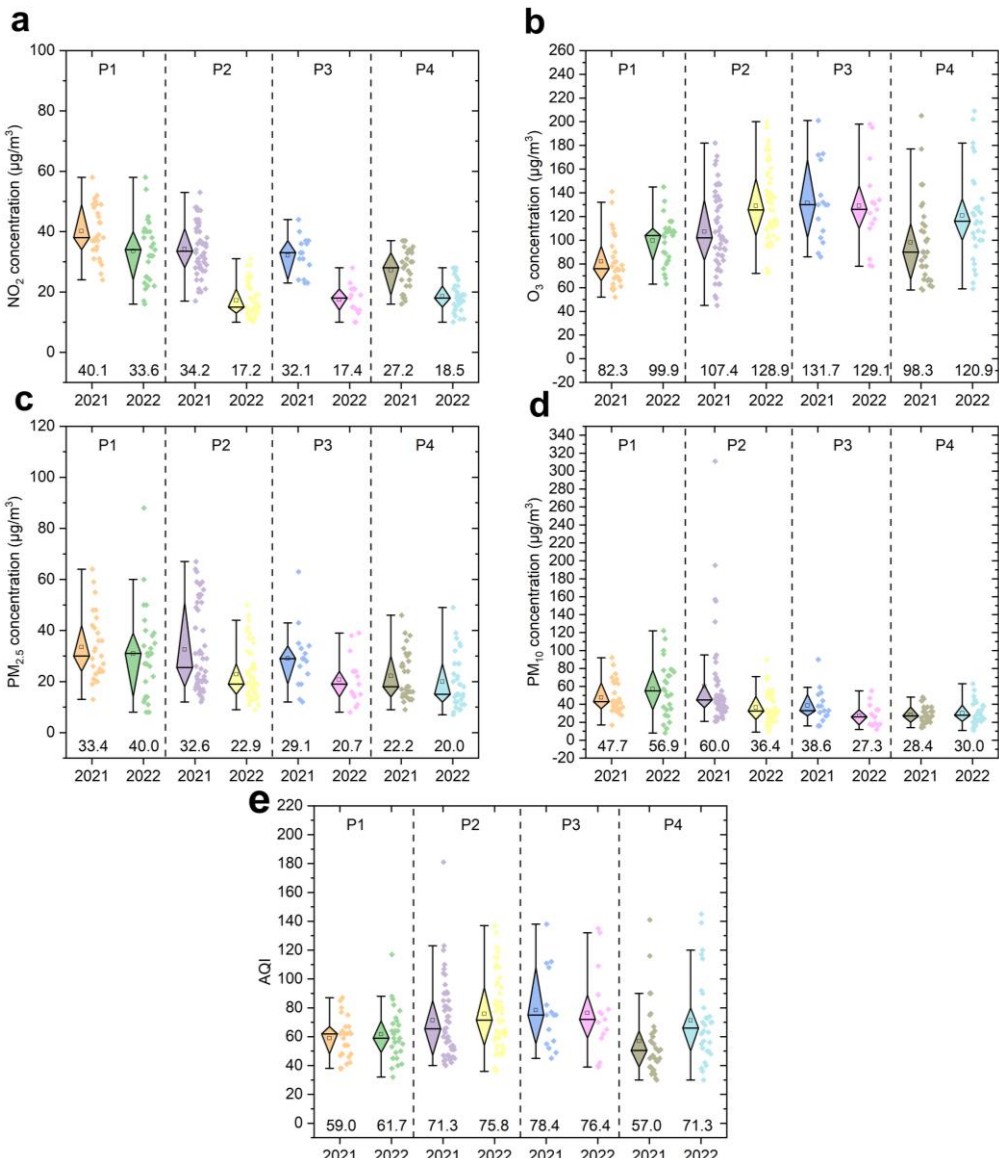

**Figure 3.** Daily air pollutant concentrations (**a**–**d**) and the AQI (**e**) during different periods (P1–P4) in 2021 and 2022 over Shanghai. The average values during each period are presented at the bottom. The horizontal bar and rectangle inside each box present the median line and the mean value, respectively.

The monthly Multiresolution Emission Inventory for China 2010 (MEIC10) shows that the major sources of $PM_{2.5}$ and $PM_{10}$ are complicated in Shanghai [61]. The anthropogenic sources of $PM_{2.5}$ and $PM_{10}$ are mainly from the industry, power generation, residential, and transportation sectors [61–63]. Thus, the overall reduction in $PM_{2.5}$ and $PM_{10}$ concentrations could have contributed to the sharp reductions in anthropogenic activities. Although the formation of secondary aerosols may result in an increase in $PM_{2.5}$ and $PM_{10}$ [64,65], there were no $PM_{2.5}$ and $PM_{10}$ pollution episodes ($PM_{2.5} > 75$ μg/m$^3$ or $PM_{10} > 150$ μg/m$^3$) observed during P2 of 2022 (Figure 2). From P3 to P4, $PM_{10}$ and $PM_{2.5}$ concentrations did not obviously rebound, despite the resumption of work and production after the lifting of lockdown restrictions (Figures 2 and 3). One possible reason could be the favorable meteorological conditions (e.g., strong solar radiation, high temperature) in June, which were conducive to PM diffusion [66]. Additionally, this result may be limited by the complex chemical composition and formation reactions of secondary aerosols, resulting in a delayed response [67].

Compared with the previous year, $NO_2$ concentrations in P2 of 2022 sharply curtailed, by 49.7% ($-17.0$ $\mu g/m^3$), with a high significant difference ($p < 0.01$) (Figure 3 and Table S5). The average $NO_2$ concentrations in P2 (17.2 $\mu g/m^3$) were significantly lower than the 40 $\mu g/m^3$ required by the EAQS level I standard. The absolute change in $NO_2$ concentrations in P2 was significantly higher than that in $PM_{2.5}$ and $PM_{10}$ concentrations, indicating that $NO_2$ had a more sensitive response to the lockdown. We further found that the freight turnover and tourist turnover (two traffic emissions indicators) in Shanghai significantly decreased in April 2022, by 34.0% and 95.7%, respectively, compared to 2021 (Figure S1). These results were mostly consistent with previous studies in Shanghai during HCWL, in which decreased $NO_2$ was strongly associated with substantial restrictions on local traffic and industrial activities [67–69]. From P3 to P4, the $NO_2$ concentrations were still significantly lower than those in 2021 ($p < 0.01$), and with the gradual recovery of human activities, $NO_2$ concentrations began to slightly recover.

The variations in the $O_3$ concentrations were more complicated. The average $O_3$ concentrations increased by 20.0% (21.5 $\mu g/m^3$) over Shanghai (Figure 3) during P2 of 2022, ranging from 72 $\mu g/m^3$ to 200 $\mu g/m^3$. According to the EAQS standard of daily MDA8 $O_3$ concentrations, 80.4% of days during P2 failed to meet the first level's standard (100 $\mu g/m^3$). What is worse, from P2 to P4 in 2022, $O_3$ concentrations remained at a high level, with the average value ranging from 120.9 $\mu g/m^3$ to 128.9 $\mu g/m^3$. More investigations into $O_3$ variations during different periods of 2022 will be further discussed in Sections 3.3 and 3.4. The AQI showed an unexpected upward trend during P2 of 2022 compared to 2021, with an enhancement of +4.5 points (6.3%), ranging from 36 to 137. This was likely due to the enhanced $O_3$ concentrations. Specifically, the proportion of $O_3$ as the dominant pollutant during P2 of 2022 increased from 62.7% to 98.0% when compared to 2021 (Figure S2), resulting in a 7.9% decrease in the good air quality rate (AQI < 100, Figure S3). Moreover, in the following periods (i.e., from P3 to P4) of 2022, the AQI was completely influenced by $O_3$ (100%, Figure S2). These results imply that the improvement in air quality from the primary emission reduction cannot completely offset the adverse effect of the increased $O_3$ during this lockdown in Shanghai.

3.1.2. Variations in Satellite-Observed Tropospheric $NO_2$ and HCHO Concentrations

In Figure 4a–h, during P1 of 2021 and 2022, the spatial distributions of the TROPOMI average $NO_2$ VCDs were similar in general. However, with the implementation of the lockdown measures in Shanghai, $NO_2$ VCDs significantly decreased during P2 of 2022, especially in the downtown and neighboring areas (Figure 4a). During P3 of 2022, $NO_2$ VCDs remained at a low level (Figure 4g), while a slight rebound occurred in P4 (Figure 4h). The variation trend for $NO_2$ observed by satellites was in good agreement with ground observations.

Figure 4m–x presents the spatial distributions and absolute differences in TROPOMI average HCHO VCDs. Temperature and solar radiation have important effects on the formation of HCHO by affecting biogenic emissions and photochemical reactions [29,70]. Additionally, the contribution from anthropogenic sources (e.g., industry and transportation) to HCHO cannot be ignored [71]. Li et al. [29] reported that HCHO VCDs in Shanghai during 2010–2019 were the highest in summer and the lowest in winter. It can be observed that from P1 to P4 in 2021, HCHO VCDs showed a typical upward trend in general (Figure 4m–p), which was significantly related to the increased temperature from spring to summer [29]. Nonetheless, from P1 to P3 in 2022, HCHO VCDs remained at a consistently low value (Figure 4q–s), suggesting that the effects of increased temperature were probably offset by the impact of the reduced anthropogenic activities. Although we have demonstrated only a sharp decrease in $NO_2$, it generally correlates to the reductions in $NO_x$ at a regional scale [70]. The reduction in anthropogenic $NO_x$ and in nonmethane volatile organic compound (NMVOC) emissions was possibly the main reason for the decline in HCHO in Shanghai [70]. During P4 of 2022, HCHO VCDs showed a greater recovery

(Figure 4t,x) caused by the warmer summer temperatures and the resumption of work and production.

### 3.2. Effect of Long-Range Transport Based on the HYSPLIT Model

Given the influences of atmospheric motion and disturbance, the long-range transport of air masses may have had an impact on the air pollutants in Shanghai. As shown in Figure 5, during P1, northerly and southerly air flows from the Yellow Sea, the East China Sea, and the coastal areas of the Bohai Sea dominated the air mass transport to Shanghai (59.87%). Similarly, during P2 and P3, four clusters of trajectories originated from the coastal waters and sea areas, accounting for 62.51% and 80.28% of the air mass transport, respectively. This indicated that the major air masses were clean and that the anthropogenic pollutant load was low.

The terrestrial inputs during P2 and P3 were mainly from the southern cities of the YRD (Figure 5b), whereas the contribution from regional transport from the northern high-polluted cities of YRD (especially Jiangsu and the central and eastern parts of Anhui) and Central China was not significant [72,73]. Therefore, anthropogenic pollutants from inland cities and industries contributed little to the substantial variations in air pollutants during the COVID-19 epidemic period in Shanghai.

We further explored the distribution characteristics of the surrounding biomass burning, such as wildfires and straw burning. According to the Landsat-8 satellite, fire spots were sparsely distributed in YRD during P2 and P3 of 2022, and no biomass burning was observed in Shanghai (Figure S4). The locations of the fire spots were also not on the 24-hour backward trajectories of Shanghai. In general, the long-range transport of land-based pollutants had little impact on Shanghai.

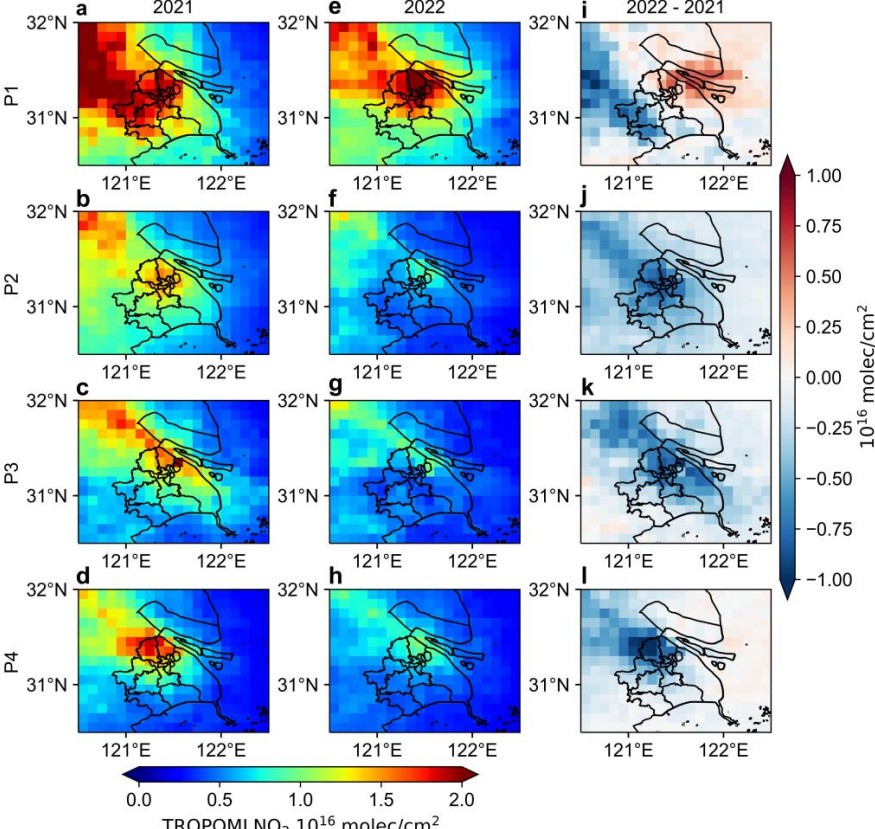

**Figure 4.** *Cont.*

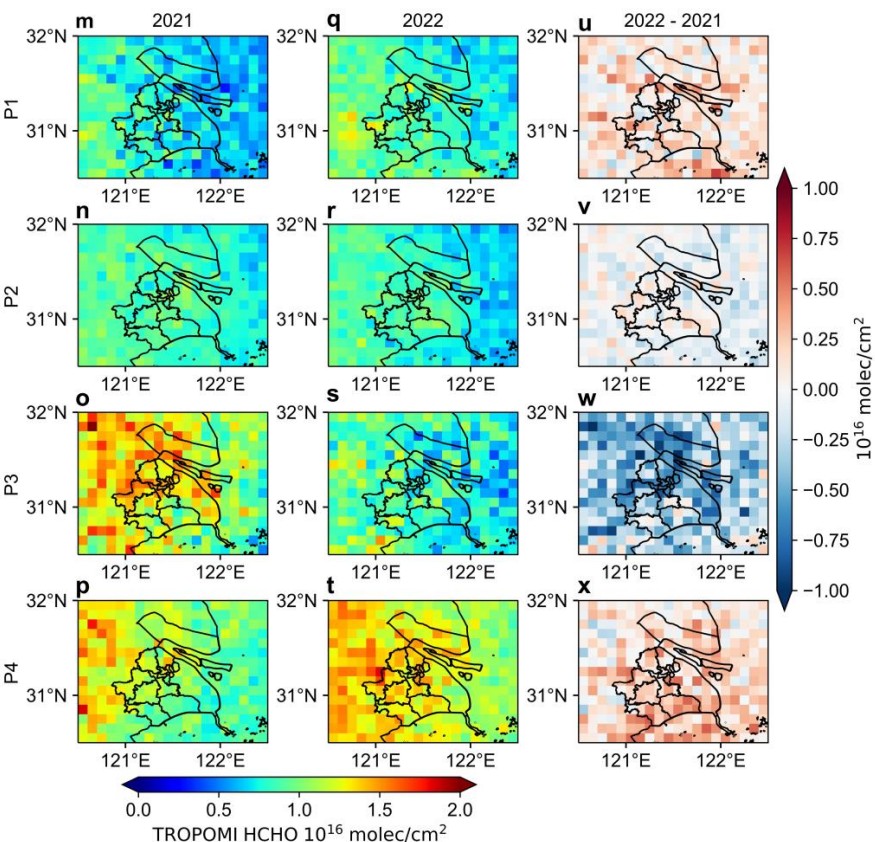

**Figure 4.** (**a–i**) Spatial distributions of TROPOMI tropospheric NO$_2$ VCDs in Shanghai from P1 to P4 in 2021 (**a–d**), 2022 (**e–h**) and the absolute differences between these two years (**i–l**); (**m–x**) is the same as (**a–i**) but for TROPOMI tropospheric HCHO VCDs.

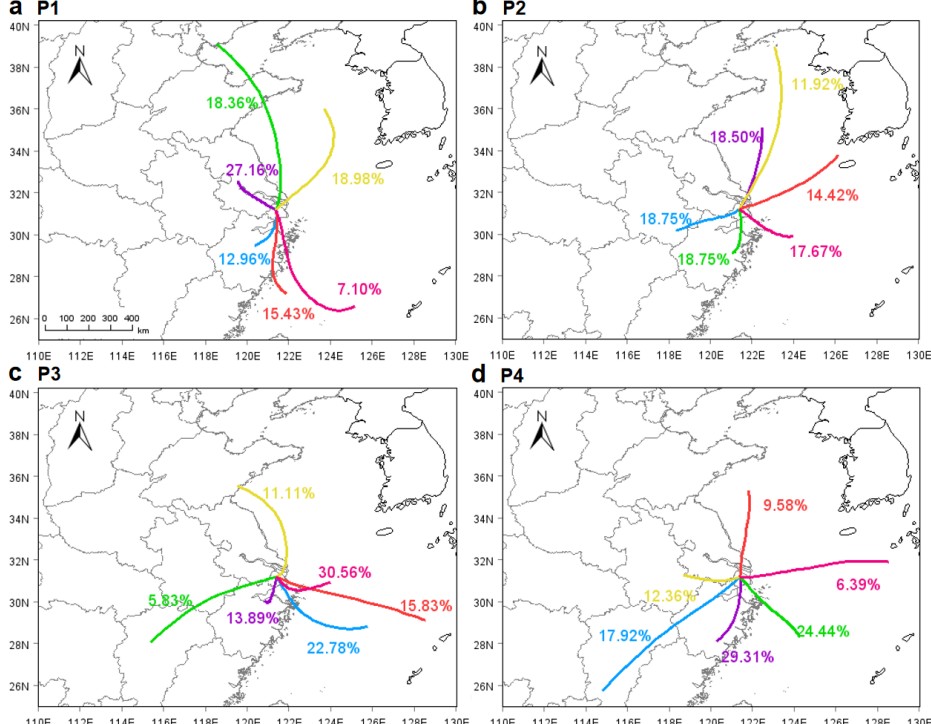

**Figure 5.** The air mass transport from P1 to P4 in 2022 (**a–d**) in Shanghai region by cluster analysis of backward trajectories.

### 3.3. Quantification of Meteorological and Anthropogenic Influences on Air Pollutants in MLR Model

By understanding the variations of different air pollutants compared with the previous year, we could conclude that anthropogenic factors have played non-negligible roles and that the long-range transport of pollutants contributed little. However, there are still questions whether these changes in air pollutants are mainly contributed by local meteorological factors or anthropogenic factors, or a combination of both, given the possible influence of anomalous weather. Therefore, we further quantified the meteorologically and anthropogenically driven changes in the $NO_2$, $O_3$, $PM_{2.5}$, and $PM_{10}$ concentrations during P2 on the basis of using the developed stepwise MLR model.

The changing trends of air pollutants in the BS site were similar to those in the whole of Shanghai (Figure S5). Compared with the 3-year baseline (2019–2021), $NO_2$, $PM_{2.5}$, and $PM_{10}$ concentrations declined by 62.2% ($-28.6$ μg/m$^3$), 34.4% ($-12.1$ μg/m$^3$), and 24.3% ($-18.7$ μg/m$^3$), respectively. In contrast, $O_3$ concentrations increased by 14.8% ($+15.6$ μg/m$^3$). According to the MLR estimation, there was no significant fluctuation in the meteorologically driven changes in air pollutants during P2 compared with that influenced by anthropogenic activities (Figure 6a–d). The total changing values of $NO_2$, $O_3$, $PM_{2.5}$, and $PM_{10}$ were $+0.9$ μg/m$^3$, $+0.6$ μg/m$^3$, $-1.2$ μg/m$^3$, and $-0.5$ μg/m$^3$, respectively. Overall, the meteorological factors slightly impacted the changes in air pollutants during P2. These findings could be verified from comparisons of the meteorological predictors during P2 of 2019–2021 and of 2022. The meteorological predictors slightly changed from the 3-year baseline during the lockdown, including a $+0.1$% change in pressure, $+1.3$% change in temperature, $-3.2$% change in wind speed, and $+0.1$% change in RH (Figure S6a–d). There is also no obvious difference in the frequency distributions of precipitation (Figure S6e,f).

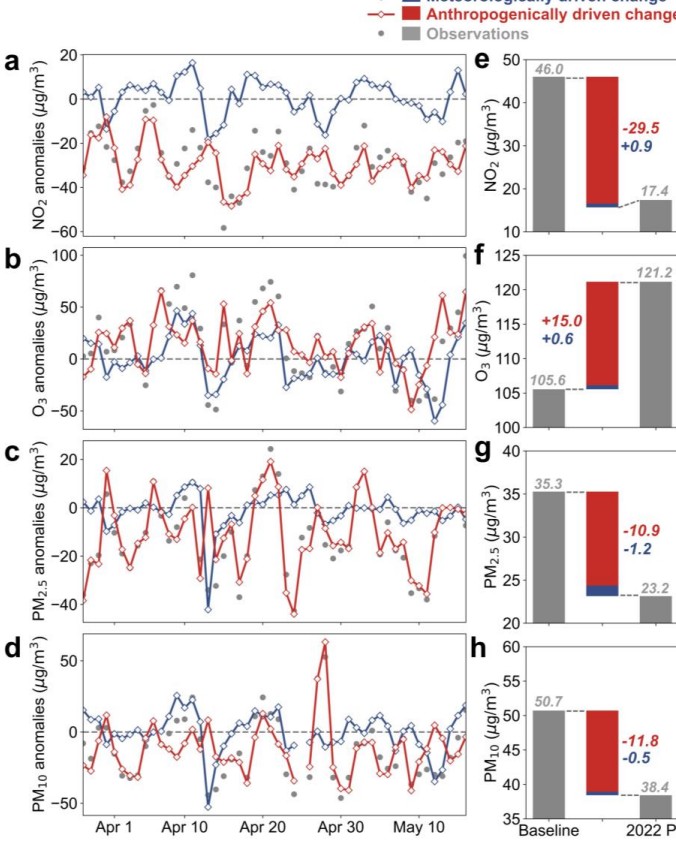

**Figure 6.** (**a–d**) Time series of meteorologically driven and anthropogenically driven air pollutant anomalies during P2 of 2022 and 3-year observed average air pollutant anomalies over BS; (**e–h**) meteorologically and anthropogenically driven changes in different air pollutants in P2 of 2022 over BS compared with the observed 3-year baseline.

The anthropogenically driven changes in $NO_2$, $O_3$, $PM_{2.5}$, and $PM_{10}$ concentrations are $-29.5 \, \mu g/m^3$ (103%), $+15.0 \, \mu g/m^3$ (96%), $-10.9 \, \mu g/m^3$ (90%), and $-11.8 \, \mu g/m^3$ (96%), respectively. This suggests that the anthropogenic factors significantly dominated the changes in all air pollutants during P2. In particular, the anthropogenically driven $NO_2$ change always presented negative anomalies during the whole lockdown (Figure 4a), ranging from $-55.2 \, \mu g/m^3$ to $-2.9 \, \mu g/m^3$. Consequently, these results provide further evidence that the $NO_2$ exposure level can be effectively sustained by reducing anthropogenic sources, such as by limiting the emission of vehicles.

### 3.4. $O_3$ Variations and Formation Regime in Different Periods

From the perspective of the monthly distribution of $O_3$ concentrations, the rate of increase in $O_3$ from March to April 2022 changed from 16.7% to 27.9% compared with the 3-year baseline (Figure 7). These also indicated the significant $O_3$ pollution in Shanghai during the lockdown. For the amplification of $O_3$ pollution, as shown in Figure 6g, anthropogenic factors were the main reason during P2. Moreover, as mentioned in Section 3.1.1, $O_3$ concentrations remained at a higher level from P2 to P4 in 2022 compared with the previous year. Owing to the adverse effect of $O_3$ on the AQI, it is urgent to explore the formation mechanism of $O_3$. The relationship between $NO_x$, VOCs, and ozone formation is nonlinear [31]. The strategy to control both $NO_x$ and VOCs on the formation of ground-level $O_3$ depends on the regional photochemical oxidation regimes, which are $NO_x$-limited regime and VOC-limited regime [31]. In brief, under a $NO_x$-limited regime, the $NO_x$/VOCs ratio is lower and the $O_3$ reduction is controlled mainly by $NO_x$, while under a VOC-limited regime, the ratio of $NO_x$/VOCs is higher and the reduction in $O_3$ is controlled mainly by VOCs. Using the criteria estimated by Duncan et al. [74], the $O_3$ formation regime is designated as a VOC-limited regime (FNR < 1), a $NO_x$-limited regime (FNR > 2), or a transitional regime (1 < FNR < 2), where $O_3$ productions can be changed by both VOCs and $NO_x$.

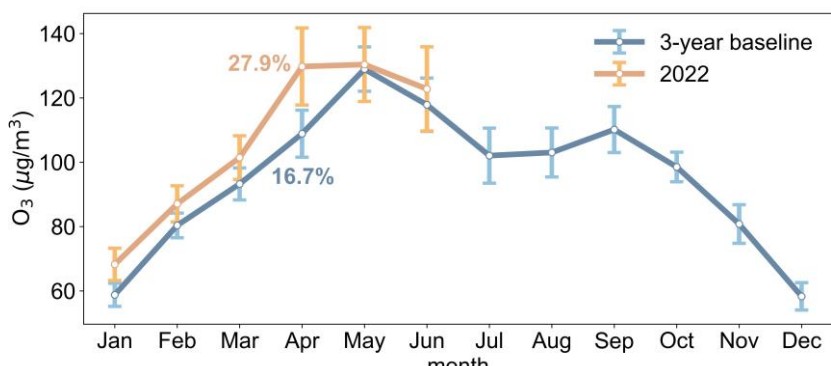

**Figure 7.** Monthly distribution of 3-year average and 2022 ground-level MDA8 $O_3$ concentrations in Shanghai, with 95% confidence intervals (error bars). The words show the rate of increase in $O_3$ concentration in April compared with that in March. Note: the 3 years are 2018, 2019, and 2021. $O_3$ in 2020 was not considered, because of the impact of the wintertime COVID-19 lockdown.

Figure 8 shows the spatial distributions of FNR in Shanghai from P1 to P4 in 2021 and 2022. During P1 and P2 of 2021, Shanghai was basically under the VOC-limited regime. From P2 to P4 in 2021 (spring to summer), adequate sunlight and more precipitation facilitated the photochemical removal and wet deposition of $NO_2$ [29]. With the decrease in $NO_2$ and the increase in HCHO (Figure 4), the $O_3$ formation regime in Shanghai mostly transformed from a VOC-limited regime to a transitional regime (Figure 8). Nonetheless, the downtown and some suburban areas (e.g., BS and JD) were more likely to be under a VOC-limited regime. Like P1 in 2021, Shanghai was controlled mainly by a VOC-limited regime during P1 of 2022. Thus, the substantial drop in $NO_x$ emissions from P1 to P2 led to a lower $O_3$ titration by NO, resulting in an increase in $O_3$ concentrations [18,20,75].

Meanwhile, the MDA8 $O_3$ concentrations in Shanghai generally present a high level from April to September (Figure 7), which is due to the strong solar irradiation and high temperature [23,76]. Therefore, the worse air quality was caused mainly by the chemical process of $O_3$ enhancement under high background $O_3$ concentrations.

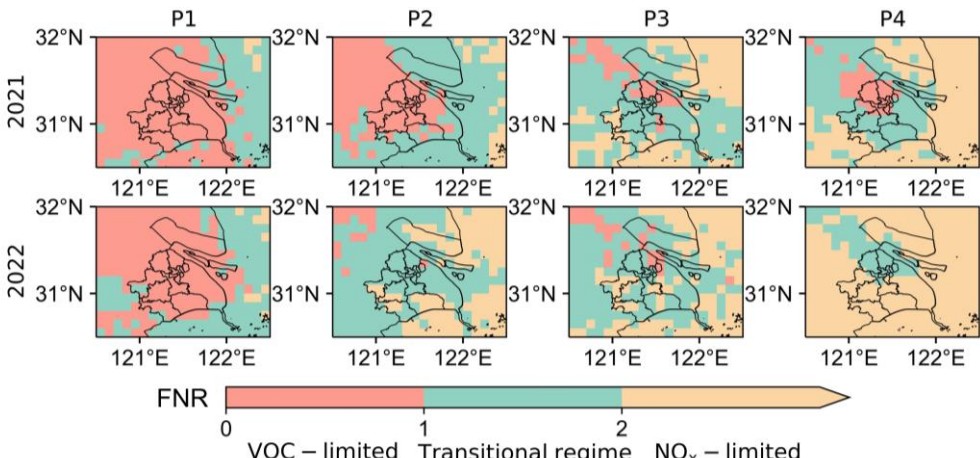

**Figure 8.** The spatial distributions of FNR in Shanghai from P1 to P4 in 2021 and 2022.

During P2 and P3 of 2022, Shanghai generally changed from a VOC-limited regime to a transitional regime caused by the significant reduction in $NO_2$ emissions. During P4 of 2022, with the significant increase in HCHO concentrations in summer and the low value of $NO_2$ (Figure 4), FNR increased, and transitional regime mostly turned into a $NO_x$-limited regime (Figure 8), especially in the southern part of Shanghai. Thus, during P3 and P4 of 2022, average $O_3$ concentrations remained at a high level (>120 µg/m$^3$) over Shanghai, which may be partially due to the effects of the transformation in the $O_3$ formation regime [34]. Additionally, the subsequent rebound of $NO_x$ may even amplify the $O_3$ pollution and worsen air quality under the $NO_x$-limited regime after P4.

### 3.5. Air Pollutants Responses to the Lockdown Measures in Downtown vs. Suburbs

Given that there are obvious spatial variances in the changes in tropospheric air pollutants (Figure 4c,f), we further quantified the interperiod absolute and relative changes of different ground-level air pollutants between downtown and the suburbs. Corresponding to the adjustment to lockdown measures, the changes from P1 to P2 (P2 minus P1) and from P3 to P4 (P4 minus P3) in 2022 were analyzed and compared with those in 2021 (Figure 9). The differences in changes in the ground-level $PM_{2.5}$ and $PM_{10}$ concentrations between downtown and the suburbs were basically the same as those in the previous year but were significant in 2022 (Figure 9a–d). Specifically, in the downtown areas, we found that the change in $PM_{2.5}$ concentrations between P1 and P2 was −18.5 µg/m$^3$ (−34.6%) in 2022, which was 5.0 µg/m$^3$ lower than that in the suburbs. However, between P3 and P4 in the downtown areas, the change in $PM_{2.5}$ concentrations was +1.0 µg/m$^3$ (+5.3%), which was 1.9 µg/m$^3$ higher than that in the suburbs. The pattern of changes in $PM_{10}$ and $NO_2$ concentrations between downtown and the suburbs was similar to that in $PM_{2.5}$ (Figure 9a–e), whereas there was no significant difference in $O_3$ changes (Figure 9f). Overall, the responses of $PM_{2.5}$, $PM_{10}$, and $NO_2$ to the changes in lockdown measures were more sensitive in the downtown areas than in the suburbs.

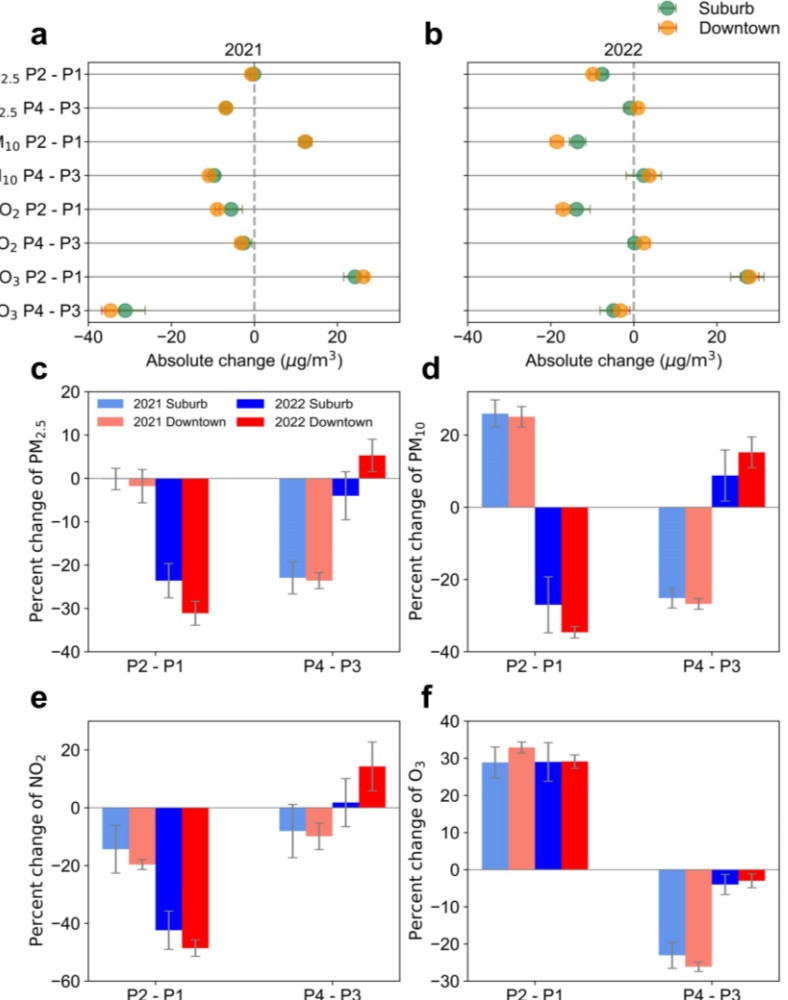

**Figure 9.** Interperiod absolute change (**a**,**b**) and relative change (**c**–**f**) in different ground-based air pollutant concentrations during 2021 and 2022 from P1 to P4 in suburbs and downtown with 95% confidence intervals (error bars).

These results are associated with regional differences. The downtown of Shanghai has a high population density of 23,091 persons per km$^2$, whereas in the suburbs, the population density is approximately eight times lower (3006 persons per km$^2$) [35]. Moreover, the traffic and road density in downtown areas is high, and heavy industries are concentrated mainly in the suburbs. For both $PM_{2.5}$ and $PM_{10}$, their sources are more strongly linked with vehicular emissions in the downtown areas [77], whereas in the suburbs, the main sources are industrial emissions and coal combustion [63,78]. For $NO_2$, many studies have suggested that $NO_2$ concentrations in Shanghai show a significant positive correlation with residence intensity, traffic, and industrial density [79]. Therefore, the larger declines or recovery in the population movements and transport activities in downtown led to more-significant changes in $PM_{2.5}$, $PM_{10}$, and $NO_2$ compared with the suburbs. Despite the higher industrial density in the suburbs, to ensure the basic operation of Shanghai and the stability of the industrial supply chain, there were still parts of necessary industrial enterprises maintaining a closed-loop production state (e.g., energy, chemicals, and electric power) [80]. This may partly explain the less-obvious changes in $PM_{2.5}$, $PM_{10}$, and $NO_2$ from P1 to P2 in the suburbs than those in the downtown areas. In terms of quarantine measures, the lockdown measures were enforced more strongly in the downtown areas because they contained the highest number of confirmed infections, which also led to a greater reduction in primary emissions compared with the suburbs. Given the highly sensitivity of $PM_{2.5}$, $PM_{10}$, and $NO_2$ in the downtown to policy restrictions on anthropogenic activities (e.g.,

transportation), future mitigating strategies for air pollution can focus more on downtown areas.

## 4. Discussion

This study revealed that the increased AQI during the COVID-19 lockdown in the spring of 2022 was influenced mainly by the enhancement of $O_3$ pollution. We further explored the influences of long-range transport and quantified the impact of meteorological and anthropogenic factors, and the results showed that the anthropogenic influence dominated the observed changes in ground-level air pollutants. The Shanghai region was controlled mainly by the VOC-limited regime during the pre-lockdown period, and thus, the substantial decline in $NO_x$ caused by reduced human activities led to increased $O_3$ formation during the full-lockdown period. Combined with the increased background $O_3$ concentrations, the $O_3$ level was high and dominated the increase in the AQI.

According to our review of the HWCL over Shanghai, many previous studies have shown that the $O_3$ concentrations had increased, whereas the air quality still improved (Figure S7) [13,18], which was different from our findings. This may be because the background $O_3$ concentrations were lowest in winter (Figure 7), and the improvement in air quality was influenced mostly by the declines in $PM_{2.5}$ and $NO_2$ during the HWCL (Figures S7 and S8). In addition, comparing our study with previous studies on the $O_3$ formation regime, we found that the VOC-limited regime was dominant before and after the HWCL in Shanghai [81]. This may be because in the winner, HCHO VCDs are at their lowest level and $NO_2$ VCDs are at their highest, resulting in the lowest FNR [29,81]. Thus, although $NO_2$ significantly declined caused by the lockdown, FNR was still within the range of less than 1 (VOC limited) [81]. Therefore, these results and comparisons indicated that anthropogenic emission reductions occurring in different seasons may lead to different air pollution mechanisms and processes.

The results of this study have some limitations. The criteria of the ozone formation regime classification for Shanghai and the FNR calculated by satellite observations may have some biases. Given the reference role of the COVID-19 lockdown measures for future air pollution control policies, future work can further focus on understanding the impact of primary aerosol changes on secondary aerosols by using regional air quality models.

## 5. Conclusions

In this work, we used the ground-based and Sentinel-5P/TROPOMI measurements to explore the variations in air quality during the new round of COVID-19 lockdowns over Shanghai in 2022. A deterioration in air quality in Shanghai during the COVID-19 lockdown in the spring of 2022 was observed. We then quantified the influence of meteorological and anthropogenic factors on the observed changes in different air pollutants and analyzed the mechanism of $O_3$ formation. The responses of air pollutants to the change in lockdown measures between downtown and the suburbs were also explored.

During the full-lockdown period, ground-level $PM_{2.5}$, $PM_{10}$, and $NO_2$ concentrations in Shanghai significantly decreased, by 29.8% ($-9.7$ μg/m$^3$), 39.3% ($-23.6$ μg/m$^3$), and 49.7% ($-17.0$ μg/m$^3$), compared with the values from the previous year, whereas $O_3$ concentrations increased by 20.0% ($+21.5$ μg/m$^3$). The AQI, the degree of air cleanliness and the impact on health, showed an upward trend, increasing by 6.3% ($+4.5$ points), which was due mainly to the amplification of $O_3$ pollution. Overall, the improvement in air quality by primary emission reduction cannot offset the adverse effect of $O_3$ pollution amplification in Shanghai. The variation trend of tropospheric $NO_2$ VCDs was highly consistent with ground-based observations. During the post-lockdown period, $PM_{10}$ and $PM_{2.5}$ in Shanghai did not significantly rebound. $NO_2$ slightly increased, while $O_3$ concentrations remained at a high level.

Using the HYSPLIT model, the backward trajectory analysis showed that the long-range transport of land-based pollutants had little impact on Shanghai. According to the MLR model, the anthropogenically driven change accounted for 103%, 96%, 90%, and

96% in the observed changes for $NO_2$, $O_3$, $PM_{2.5}$, and $PM_{10}$ concentrations during the full-lockdown period, respectively, whereas the meteorological factors contributed little ($-3\sim10\%$). Moreover, the anthropogenically driven $NO_2$ change always presented negative anomalies, further indicating that $NO_2$ was the most sensitive to anthropogenic activities.

The amplification of $O_3$ pollution adversely affected the AQI during the full-lockdown period. One of the reasons was that under the VOC-limited regime, the substantial drop in $NO_x$ emissions led to an increase in $O_3$ concentrations. Another reason was the general increase in background $O_3$ concentrations was due to seasonal variations. However, during the post-lockdown period, the $O_3$ formation regime transformed into a $NO_x$-limited regime because of the large reduction in $NO_2$ emissions and significantly increased HCHO (an indicator of VOCs) in summer. Under the $NO_x$-limited regime, the subsequent rebound of $NO_x$ may amplify $O_3$ pollution and even make air quality worse.

The responses of $PM_{2.5}$, $PM_{10}$, and $NO_2$ to full-lockdown measures and the post-lockdown measures in downtown were all more sensitive than those in the suburbs. There was no significant difference in $O_3$ changes between downtown and the suburbs.

This study facilitated the understanding of air quality responses during the COVID-19 lockdowns. Despite the substantial reductions in the primary emissions, air quality was not improved in Shanghai, which revealed the important roles of the $O_3$ formation mechanism. Overall, the different seasons during which COVID-19 lockdowns occurred may have led to different side effects on the air pollution process, which need to be carefully considered in air pollution control strategies.

**Supplementary Materials:** The following supporting information can be downloaded at https://www.mdpi.com/article/10.3390/rs15051295/s1, Text S1–S4: the supplementary information on the AQI, IAQI, FNR, Wilcoxon Signed Rank Test, and the TROPOMI $NO_2$ and HCHO tropospheric VCDs data; Figures S1–S8: the supplementary figures about air pollution and meteorology in Shanghai; Tables S1–S8: the supplementary summary of research datesets and the detailed results of the normal distribution test, MLR model analysis, and Wilcoxon Signed Rank Test [82–85].

**Author Contributions:** Conceptualization, L.Z.; methodology, Q.M.; formal analysis, Q.M.; investigation, Q.M.; data curation, Q.M., J.W., and M.X.; writing—original draft preparation, Q.M., J.W., and M.X.; writing—review and editing, L.Z.; visualization, Q.M., J.W., and M.X.; supervision, L.Z. All authors have read and agreed to the published version of the manuscript.

**Funding:** This research was funded by the National Natural Science Foundation of China grant no. 41975139 and the National Key R&D Program of China (2019YFC1510400).

**Data Availability Statement:** The AQI, daily averaged concentrations of $PM_{2.5}$, $PM_{10}$, and $NO_2$, daily MDA8 $O_3$ concentrations, and daily dominant pollutant can be obtained from the Shanghai Municipal Ecological Environment Bureau (https://sthj.sh.gov.cn/, accessed on 10 December 2022). The meteorological data at the BS meteorological station can be obtained from the National Climatic Data Center (ftp://ftp.ncdc.noaa.gov/pub/data/noaa/isd-lite/, accessed on 10 December 2022) and the UK Meteorological Office (http://rp5.ru/, accessed on 10 December 2022). The TROPOMI tropospheric L2 $NO_2$ and HCHO OFFL data can be obtained from the Copernicus Open Access Hub (https://s5phub.copernicus.eu/dhus/#/home, accessed on 10 December 2022).

**Acknowledgments:** The authors first gratefully thank all the anti-epidemic workers in Shanghai for their selfless effort and devotion. We acknowledge the Shanghai Municipal Ecological Environment Bureau, the National Climatic Data Center, the UK Meteorological Office, and NASA for their efforts in making the data available.

**Conflicts of Interest:** The authors declare no conflict of interest.

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
