# Peer review of "Air Quality Index (AQI) Did Not Improve during the COVID-19 Lockdown in Shanghai, China, in 2022, Based on Ground and TROPOMI Observations"

_remotesensing, doi:10.3390/rs15051295_

Round 1

Reviewer 2 Report

    This manuscript studied the density of anthropogenic air pollutants in Shanghai before, during, and after the lockdown period of Covid-19 between 2019 and 2022. It monitors Air Quality Index (AQI) based on the measurement of the trace gases concentration from ground stations and satellite sensors. It also analyzed the statistical climate data to find their impact on the AQI. A correlation and validation have been made between the density of pollutants and lockdown periods. Results show that the AQI increases (become worst) during the lockdown period. The density of O3 and PM has increased significantly, and the meteorological condition does not impact the AQI much. In addition, authors have found that O3 dominates where the concentration of NO2 decreases. Finally, the PM is more sensitive during the complete lockdown in rural areas (tows). This contribution is an actual case study useful for many remote-sensing Earth observation communities. Generally, the paper’s architecture is acceptable; however, some parts should be improved by adding details related to remote sensing techniques, especially in sections “2. Method”.

    The introduction is straightforward, providing background on air pollution and its negative impact on humans and the environment, notably in Shanghai. A general review of Covid-19 was included in the introduction to explain the effects of the total lockdown on usual activities in China. Besides, many related works were well presented concerning air pollution monitoring. Practical methods and information were introduced in the material, notably the studied zone and data sources. The result section is clear and rich in correlation graphs and maps. However, the conclusion is clear could still be improved.

    I would suggest a few revisions:

·       In 2.2.1. section, you could have a table summarizing the data sources (Spatial and temporal resolutions, variables (O3, NO2, etc.), unit, and so on. ;).

·       It is also recommended to include the Carbon Monoxide (CO) using MetOp (IASI, GOME-2) data and CH4 from AQUA (MLS) and using the FengYun data.

·       What is the vertical altitude of the measurement of VCD?

·       Have you validated in situ data with satellite results?

·       Is Figure 4 show the average (Aggregation) of trace gases over 1 year?

·       Please add a small paragraph (2 to 3 lines) before each beginning of the sections.

For example, 3. Result

This section shows the attained results …

You could also refer to some useful references discussing the monitoring g of air quality using satellites.

https://www.sciencedirect.com/science/article/pii/S2352938522000659

https://doi.org/10.1007/978-3-030-90633-7_719.4.

doi: 10.1016/j.compeleceng.2021.107257 

    In brief, the paper needs minor changes. Thus, the content also needs some enhancement. Therefore, I encourage the publication of this manuscript in “Remote Sensing.

Round 2

Reviewer 1 Report

The paper «Air quality not improved during the COVID-19 lockdown over Shanghai, China in 2022 based on ground and TROPOMI observations» has been improved and supplemented with useful information.

All of my remarks and comments were taken into account.

For version 2:

Minor remarks (not a critical)

1. “2.2.2 Sentinel-5P/TROPOMI NO2 and HCHO data”, L 207-209;

Here (or in supplementary file) please add an information about typical number of primary pixels used for averaged diurnal means for 0.1x0.1-degree grid.

2. “3.1.2 Variations of satellite observed tropospheric NO2 and HCHO”, Figure 4: Please add to figure capture an information about TROPOMI data spatial resolution (0.1x0.1-degree).

3. Comment for future research (not a critical here, i.e. not for present manuscript), if you would use TROPOMI L2 data on tropospheric NO2 and HCHO one more time.

Neither qa > 0.5 nor qa > 0.75 guarantee complete “removing cloud-covered scenes”, as was reported in (Eskes et al., 2022 and Romahn et al, 2022), as well as an absence of pixels with negative diurnal VSDs.  I’d recommend using the additional filters a) for cloud radiance fraction: 0 ≤crf≤1 and b) for VSD > 0 for every primary pixel (when you retrieve averaged VSD for different grids). If you are interested, check it out.

In present study you used averaging of VSDs on relatively long time-periods, and presence of some number of “artifact” pixels (or “pixels with errors”) in VSD distributions do not (probably) lead to significant inaccuracies.
